# Traveling waves in the human visual cortex: An MEG-EEG model-based approach

Laetitia Grabot[1,2]*, Garance Merholz[1], Jonathan Winawer[3,4], David J. Heeger[3,4], Laura Dugué[1,5]

**1** Université Paris Cité, CNRS, Integrative Neuroscience and Cognition Center, Paris, France, **2** Laboratoire des Systèmes Perceptifs, Département d'études Cognitives, École normale supérieure, PSL University, CNRS, Paris, France, **3** Department of Psychology, New York University, New York, New York, United States of America, **4** Center for Neural Science, New York University, New York, New York, United States of America, **5** Institut Universitaire de France (IUF), Paris, France

* laetitia.grabot@gmail.com

## Abstract

Brain oscillations might be traveling waves propagating in cortex. Studying their propagation within single cortical areas has mostly been restricted to invasive measurements. Their investigation in healthy humans, however, requires non-invasive recordings, such as MEG or EEG. Identifying traveling waves with these techniques is challenging because source summation, volume conduction, and low signal-to-noise ratios make it difficult to localize cortical activity from sensor responses. The difficulty is compounded by the lack of a known ground truth in traveling wave experiments. Rather than source-localizing cortical responses from sensor activity, we developed a two-part model-based neuroimaging approach: (1) The putative neural sources of a propagating oscillation were modeled within primary visual cortex (V1) via retinotopic mapping from functional MRI recordings (encoding model); and (2) the modeled sources were projected onto MEG and EEG sensors to predict the resulting signal using a biophysical head model. We tested our model by comparing its predictions against the MEG-EEG signal obtained when participants viewed visual stimuli designed to elicit either fovea-to-periphery or periphery-to-fovea traveling waves or standing waves in V1, in which ground truth cortical waves could be reasonably assumed. Correlations on within-sensor phase and amplitude relations between predicted and measured data revealed good model performance. Crucially, the model predicted sensor data more accurately when the input to the model was a traveling wave going in the stimulus direction compared to when the input was a standing wave, or a traveling wave in a different direction. Furthermore, model accuracy peaked at the spatial and temporal frequency parameters of the visual stimulation. Together, our model successfully recovers traveling wave properties in cortex when they are induced by traveling waves in stimuli. This provides a sound basis for using MEG-EEG to study endogenous traveling waves in cortex and test hypotheses related with their role in cognition.

**Data availability statement:** Modeling codes (toolbox and scripts to reproduce the main figures of the article) alongside an example dataset with instructions on how to run the codes, and the full MEG/EEG/MRI dataset are available on Zenodo (https://doi.org/10.5281/zenodo.13968952). The full dataset contains preprocessed and fully anonymized data, in compliance with ethical regulations. The latest code can be found on GitHub at https://github.com/LaetitiaG/wavesmodel.

**Funding:** This project has received funding from the European Research Council (ERC) under the European Union's Horizon 2020 research and innovation programme (grant agreement No 852139, to LD, https://cordis.europa.eu/project/id/852139). The funder had no role in study design, data collection and analysis, decision to publish, or preparation of the manuscript.

**Competing interests:** The authors have declared that no competing interests exist.

## Author summary

Brain oscillations, thought to be crucial for many cognitive processes, might actually be waves that travel across the brain's surface. Understanding these traveling waves is notoriously difficult because current non-invasive methods like magneto- and electro-encephalography (MEG-EEG) face significant technical limitations. To address this challenge, we developed a new approach that combines brain imaging techniques and computational modeling. We focused on the primary visual cortical area (V1) of the brain and created a model that simulates traveling activity across the cortex and predicts how these traveling waves should appear in EEG and MEG recordings. We tested our model by comparing its predictions with brain data collected when participants view visual patterns specifically designed to induce traveling waves in the visual system. The results show that our model accurately captures the direction and pattern of the traveling waves, as well as the specific parameters of the visual stimuli. This novel modeling toolbox offers a promising method for studying endogenous traveling waves and will enable neuroscientists to explore hypotheses about the spatiotemporal organization of brain activity and its role in cognition.

## Introduction

Neural oscillations are observed across many species at all spatial scales, from extracellular to large-scale magneto- and electro-encephalography (MEG-EEG) recordings. These oscillations affect cognitive functions such as perception, memory and attention [1–4]. This body of work usually focused on the temporal dynamics of the oscillations, characterized by their frequency and phase. Their spatial dynamics, or how the oscillations travel across the cortex, are understudied, even though they could be key to uncovering the role of oscillations in cognition, particularly with regard to synchronization across neuronal assemblies [5,6]. This gap is partly due to technical limitations, as studying traveling waves requires both high temporal and spatial resolution [7], and non-invasive measures of human brain activity typically have high resolution in only one domain or the other. We aimed to address this gap by developing a model-based neuroimaging approach to characterize traveling waves within a given brain area using MEG-EEG in humans.

Traveling waves are ubiquitous in the brain [8], and observed in many frequency bands [9–14] (e.g., alpha [9–10], theta [11], beta [12,13], gamma [14]).They have been related to different cognitive processes such as predictive coding [15,16], memory [11,17] and visual processing [18–20]. One usually distinguishes global (or macroscopic) traveling waves, observed across the cortex, from local (or mesoscopic) traveling waves, recorded within a given brain area. Global traveling waves were observed in humans using both invasive [7,9,11,17] and non-invasive techniques [15,16,21]. Local traveling waves were observed in many different brain regions (V1: 22–23, MT: 24, M1: 13, V4: 18) [13,18,22–24], but only using invasive techniques, or indirectly using behavioral measures [19,20].

There are currently no non-invasive MEG-EEG studies investigating the local propagation of neural oscillations in humans. Achieving this would require recording oscillations within a specific brain area with sufficient precision to assess their spatial propagation. However, EEG and MEG record the electrical and magnetic brain activity, respectively, with a high temporal resolution (<1ms) but poor spatial resolution. Even more detrimental for the analysis of the spatial distribution of brain dynamics, each MEG-EEG sensor captures signals from different neuronal populations, potentially far from each other, resulting in source mixing and spurious coupling [25]. Authors have inferred the putative sources of brain activity from signals recorded by sensors on the scalp by attempting to solve the inverse problem. There is, however, no unique mathematical solution for source estimation. To circumvent these issues, we propose a model-based approach using an encoding model [26,27] able to assess local traveling waves.

Our two-part model (1) simulates traveling waves in the primary visual area (V1), and (2) projects this source activity into the sensor space (Fig 1A) to be compared with actual recordings. We simulated brain activity in V1 as a proof-of-concept of our new method, but one can in principle simulate activity in any area with topographic organization. A simultaneously recorded MEG and EEG dataset was acquired to validate the model. Participants were exposed to full-screen visual stimuli carefully designed to elicit traveling waves in V1 (Fig 2), with three types of waves tested: standing, traveling from the periphery to the fovea ("traveling in") or from the fovea to the periphery ("traveling out", Fig 1B). The traveling waves were 5-Hz oscillations with a monotonic phase shift across eccentricities, while the standing waves were identical

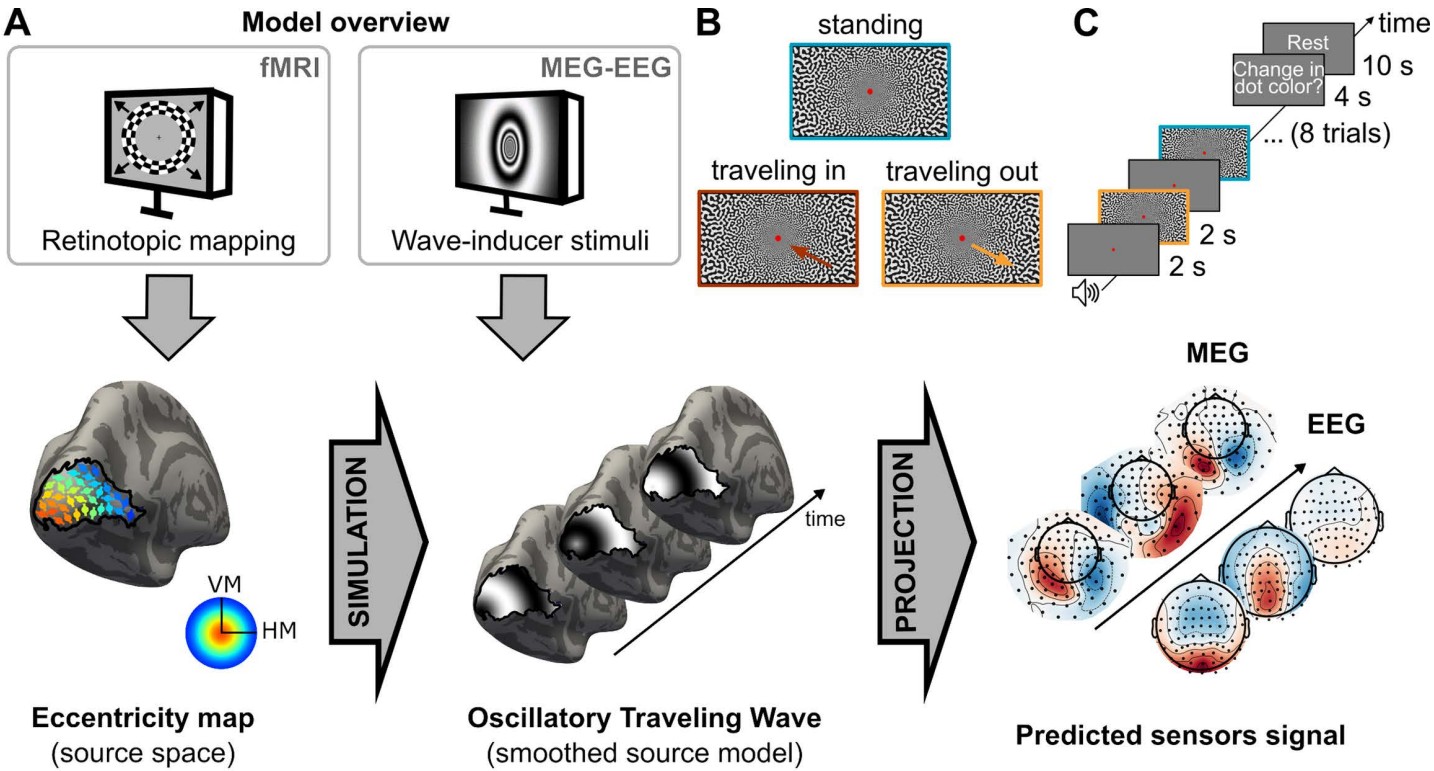

**Fig 1. Model overview. A.** The model predicts brain activity in MEG and EEG sensors based on simulations of propagating activity in primary visual area (right and left V1). First, brain activity is modeled in V1 using participants' individual retinotopic maps obtained with anatomical and functional MRI. Second, this activity is projected onto MEG and EEG sensors. **B.** To test the model, we recorded simultaneous MEG-EEG activity when participants were presented with stimuli designed to elicit specific patterns in V1. There were three types of stimuli: traveling out, traveling in and standing. **C.** Experimental protocol. Each stimulus was presented for 2 s, following a 2s-grey screen (8 trials per run). At the end of each run, participants were asked to report if they had seen the fixation point change color. VM: vertical meridian. HM: horizontal meridian.

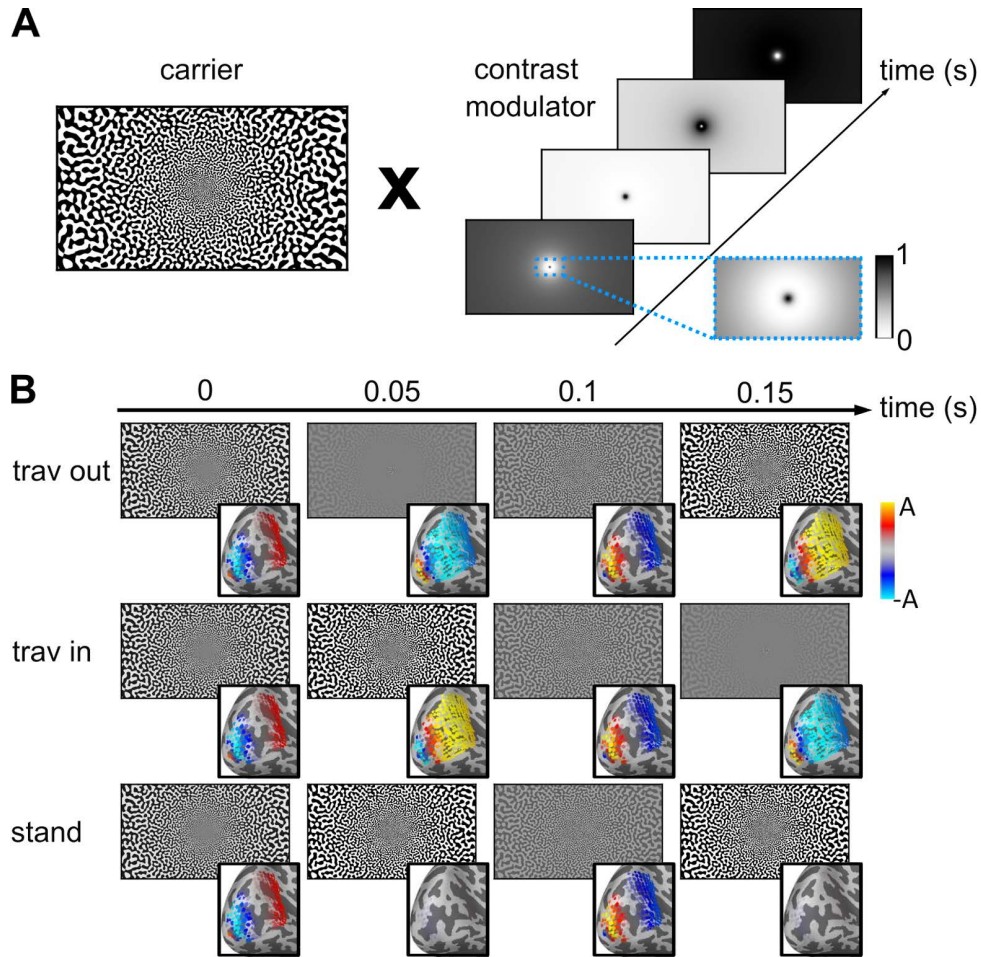

**Fig 2. Visual stimuli designed to induce traveling waves in V1. A.** Each stimulus corresponds to the multiplication of a static, black and white pattern (carrier) with a contrast modulator. The modulator is adjusted for cortical magnification, i.e., higher spatial frequency at the fovea (see zoomed area in blue). **B.** Example stimuli presented at four time points for each condition (rows: traveling out, traveling in, standing), with the corresponding patterns simulated in V1, used as model inputs (bottom right insets). A is the maximal amplitude of the simulated signal (A=0.5).

5-Hz oscillations with no spatial phase shift. Retinotopic maps of individual participants were also obtained using anatomical and functional MRI (fMRI), and used to map the visual stimuli onto V1. Measured data were then compared to the signals predicted by the model. By capturing phase and amplitude relations between sensors, our method was able to discriminate traveling waves from standing waves, and the correct direction of traveling waves from the opposite direction. We were also able to accurately recover the parameters (spatial and temporal frequencies) used in the visual stimulation, validating the method. This work opens a new avenue for studying how neural oscillations unfold through time and space using non-invasive recordings.

## Results

### The model captured phase and amplitude relations between sensors

We presented participants with visual stimuli designed to trigger traveling or standing waves at 5 Hz in V1 (Fig 2) while their brain activity was recorded with concurrent MEG and EEG. We modeled the neural responses in each participant's V1 from each of the three 5-Hz stimulus types (traveling out, traveling in, or standing wave; S1 Video),

then projected this source activity to the MEG-EEG sensors (Figs 1 and 2B and S2 Video). Oscillatory activity was measured at the 5-Hz-stimulation frequency (and harmonics) in the sensor evoked responses and power spectral densities for all conditions (S1 Fig). The three types of waves elicited different time courses, evident both in the data (Fig 3, top) and the model predictions (Fig 3, bottom) for both MEG (magnetometers, MAG) and EEG sensors. Note that although the largest signal was found in occipital sensors, V1 responses are expected to have some effect on all sensors including frontal ones, as shown by the topography of predicted signals (Fig 3, bottom). Visualization of measured and predicted phase and amplitude at 5 Hz is shown for each condition in Fig 4A (and S2 Fig for gradiometers, GRAD). The correlation between predicted and measured data, combined across participants, was significantly different from a null distribution obtained from shuffling pairs of sensors (all p-values < 0.0001 for each sensor type; see the average data point above the red dashed line indicating the null distribution in Figs 4B and S2). Furthermore, the correlations were significant for most individual participants (all participants for MAG and GRAD, 18 out of 19 participants for EEG).

### The model is specific to conditions and wave parameters

To test the specificity of the model, we compared the data from one condition to the model of the same condition ("match") or to the models of the other conditions ("cross"). We expected the matched comparison (i.e., standing model to standing data, traveling model to traveling data) to explain more variance than the crossed comparison (standing model to traveling data, traveling model to standing data, and all other possible combinations). Indeed, the matched models were about twice as accurate for the MAG and GRAD, explaining 44% of the variance in the data compared to 23% for the crossed models (Figs 4B and S2, Cohen's q = 0.24, small to medium effect size). For the EEG data, the matched models were about 34% more accurate, explaining 47% of the variance in the data compared to 35% (Fig 4B, Cohen's q = 0.14, small effect size). These results were obtained using a linear mixed model on the correlation coefficients with comparison (matched, crossed) and sensor type (MAG, GRAD, EEG) as predictors, and participants as random effect. It showed a

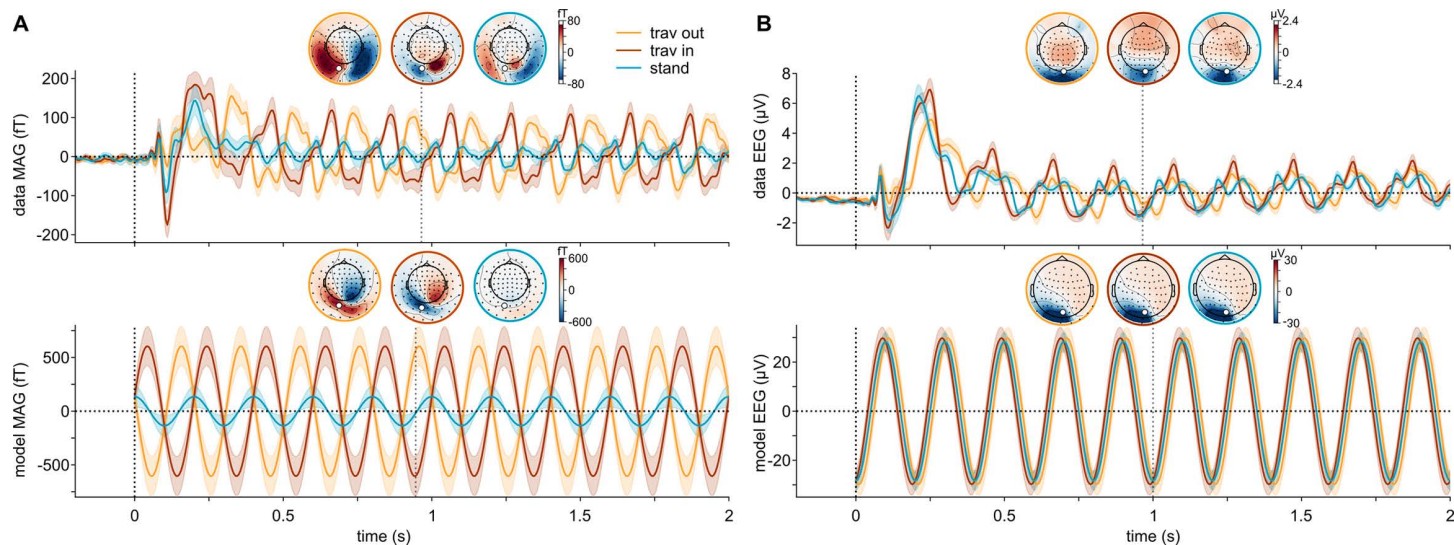

**Fig 3. Measured and predicted time courses. A.** Time courses of magnetometer (MAG) data (top) and model (bottom) for each condition (traveling out in yellow, traveling in in red and standing in blue) for an occipital sensor (highlighted in white in the topomaps). Topomaps for each condition are plotted at the time point indicated by the vertical dashed line below the topomaps. Note that amplitude values in the model are arbitrary (see Methods). **B.** Time courses of EEG signal. Same convention as in A.

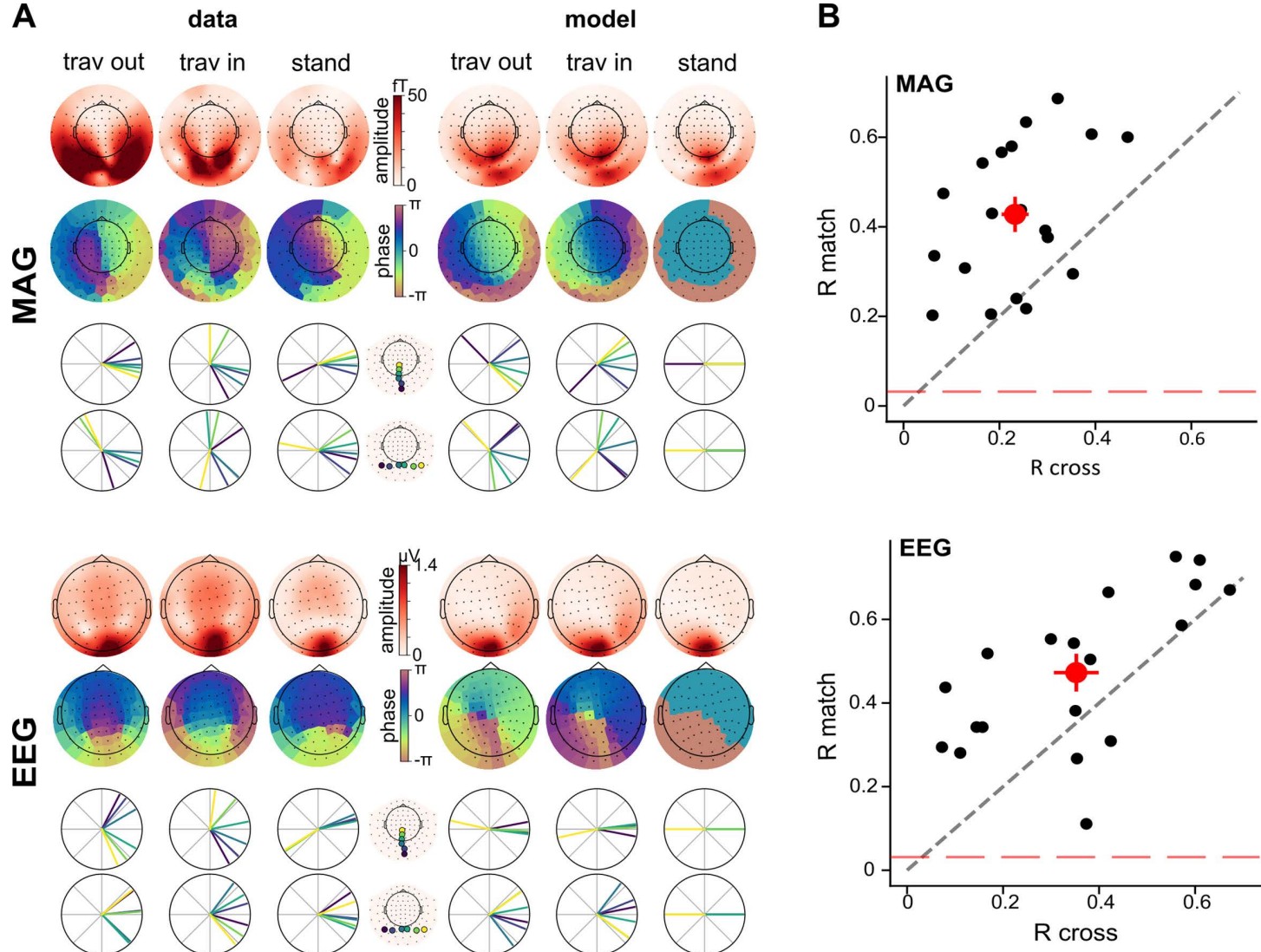

**Fig 4. The model captures phase and amplitude relations between sensors. A.** Topomaps of amplitude and phase at 5 Hz for MAG (top) and EEG sensors (bottom), for both data (left) and model (right). Polar plots show the phase of selected sensors, along horizontal and vertical meridians. **B.** All datasets are compared either with the "match" model (e.g., traveling out model for traveling out condition) or with "cross" (unmatched) models. The correlation coefficients for matched comparisons (R match) are plotted against the coefficient for crossed comparisons (R cross), for MAG (top) and EEG sensors (bottom). Each black dot is one participant. The red dot is the average across participants, error bars are standard error of the mean. Dots above the bisector (black dashed line) mean that the matched models are more accurate than the cross models. The red dashed line corresponds to the permutation-based null threshold.

main effect of comparison (F = 35.27, p = 1.28e-5), sensor type (F = 4.66, p = 0.0159) and a significant interaction between these two factors (F = 8.933, p = 7e-4). Post-hoc t-tests showed that the coefficients were stronger in EEG compared to GRAD (t = 2.5, p = 0.018) and MAG (t = 3.0, p = 4.6e-3; no significant difference between MAG and GRAD). The significant interaction and post-hoc t-tests revealed that the stronger coefficients in EEG were actually driven by stronger coefficients in the crossed condition only (crossed condition: EEG vs MAG t = 3.2, p = 0.005; EEG vs GRAD t = 3.3, p = 0.004; matched condition: EEG vs MAG t = 1.2, p = 0.256; EEG vs GRAD t = 0.3, p = 0.773; other comparisons: n.s.), indicating that EEG predictions are less specific than MEG predictions.

A long-term goal is to use this novel method to detect endogenous waves (i.e., cortical waves elicited by stimuli that do not contain traveling waves). For endogenous waves, the wave speed and direction would be unknown. Here, we used stimuli with fixed parameters to elicit fully controlled traveling waves. Hence, we asked whether we could infer the stimulus properties by blinding ourselves to the exact stimulus properties, comparing model accuracy for several combinations of spatial and temporal frequencies: temporal frequencies from 3 to 7 Hz in steps of 1 Hz, and spatial frequencies of 0.01, 0.05 and 0.1 cycles/mm. The results showed that the largest difference between the matched and crossed model was observed for the correct stimulus parameters (spatial frequency = 0.05 cy/mm and temporal frequency = 5 Hz; Figs 5A and S2C), confirming that the model is capable of distinguishing between different traveling waves. This was quantified using a linear mixed model on the correlation coefficients with comparisons (match vs. cross), sensor type, temporal frequency, and spatial frequency as factors (including the interactions with comparisons and the other factors), and participants as random effect. It showed a main effect of comparisons (F = 35.59, p = 1.21e-5), sensor type (F = 36.49, p = 2.19e-9), temporal frequency (F = 10.38, p = 1.08e-6) and spatial frequency (F = 4.74, p = 0.015). Crucially, the interactions between comparisons and temporal (F = 19.47, p = 6.82e-11) and spatial (F = 8.116, p = 0.001) frequency were significant. For each sensor type, we performed post-hoc t-tests between matched and crossed comparisons corrected for multiple comparisons (across sensors, temporal and spatial frequencies; FDR correction, Fig 5A).

### Differences of model performance across conditions

As this method is intended to contrast different models in order to test specific hypotheses regarding traveling waves, we investigated whether the model performs differently across the different stimulus conditions. We compared the matched to the crossed comparison for each measured dataset separately (traveling in, traveling out, standing; Figs 5B and S2D). For MAG and GRAD, we found that the matched model explained more variance than the crossed model for traveling out and traveling in datasets, but not for the standing data. In other words, the standing model was not better than the traveling models to predict the measured data elicited by the standing stimuli. For EEG sensors, the matched model explained more variance than the crossed model for traveling in and standing datasets, but not for the traveling out data. Such differential effect between EEG and MEG model quality is reflective of intrinsic differences between the two recording techniques. These results were obtained using a linear mixed model on the correlation coefficients with comparisons, sensor type, dataset as factors (including the interactions with comparisons and the other factors), and participants as random effect. The main effects and the interaction were significant (comparison: F = 11.53, p = 0.003; dataset: F = 3.74, p = 0.033; interaction: F = 4.206, p = 0.023). Overall, the correlation coefficients were thus higher in matched versus crossed comparison, as found in the previous analysis which concatenated all conditions together. When considering separately the different measured datasets, post-hoc t-tests show that correlation coefficients are significantly higher for matched than crossed models only for the traveling in and traveling out datasets, not for the standing dataset (traveling out: t = 4.9 p = 8.7e-6, traveling in: t = 6.4, p = 3.1e-8, standing: t = 1.0, p = 0.312). The sensor type was also a significant main effect (F = 10.8, p = 2.1e-4), with a significant triple interaction with comparison and dataset (F = 4.852, p = 0.002). This interaction reveals that, for MEG sensors, the difference between matched and crossed comparisons is significant for traveling in (FDR-corrected t-tests; MAG: t = 3.7, p = 0.005; GRAD: t = 4.9, p = 6.8e-4) and traveling out (MAG: t = 2.8, p = 0.023; GRAD: t = 4.8, p = 6.8e-4), but not for the standing condition (MAG: t = -0.4, p = 0.696; GRAD: t = 0.416, p = 0.696), while a different pattern is observed for EEG: the difference here is significant for traveling in (t = 2.7, p = 0.023) and standing (t = 3.2, p = 0.012), but not traveling out (t = 1.3, p = 0.261).

One contributing factor to the differences observed between conditions is that the different model predictions are not equally discriminable from each other. All models simulate 5Hz oscillations that may show similar amplitude or phase at given sensors, even if the overall topographic distribution differs between models. To estimate the similarity among model predictions, we calculated the correlation across sensors for each pair of model predictions: the higher the correlation, the more similar, and the less discriminable, the predictions are. The correlation coefficients were low, meaning high discriminability for traveling out versus in (MAG: 0.56 ± 0.26, GRAD: 0.50 ± 0.25, EEG: 0.53 ± 0.25, mean ± std) compared with

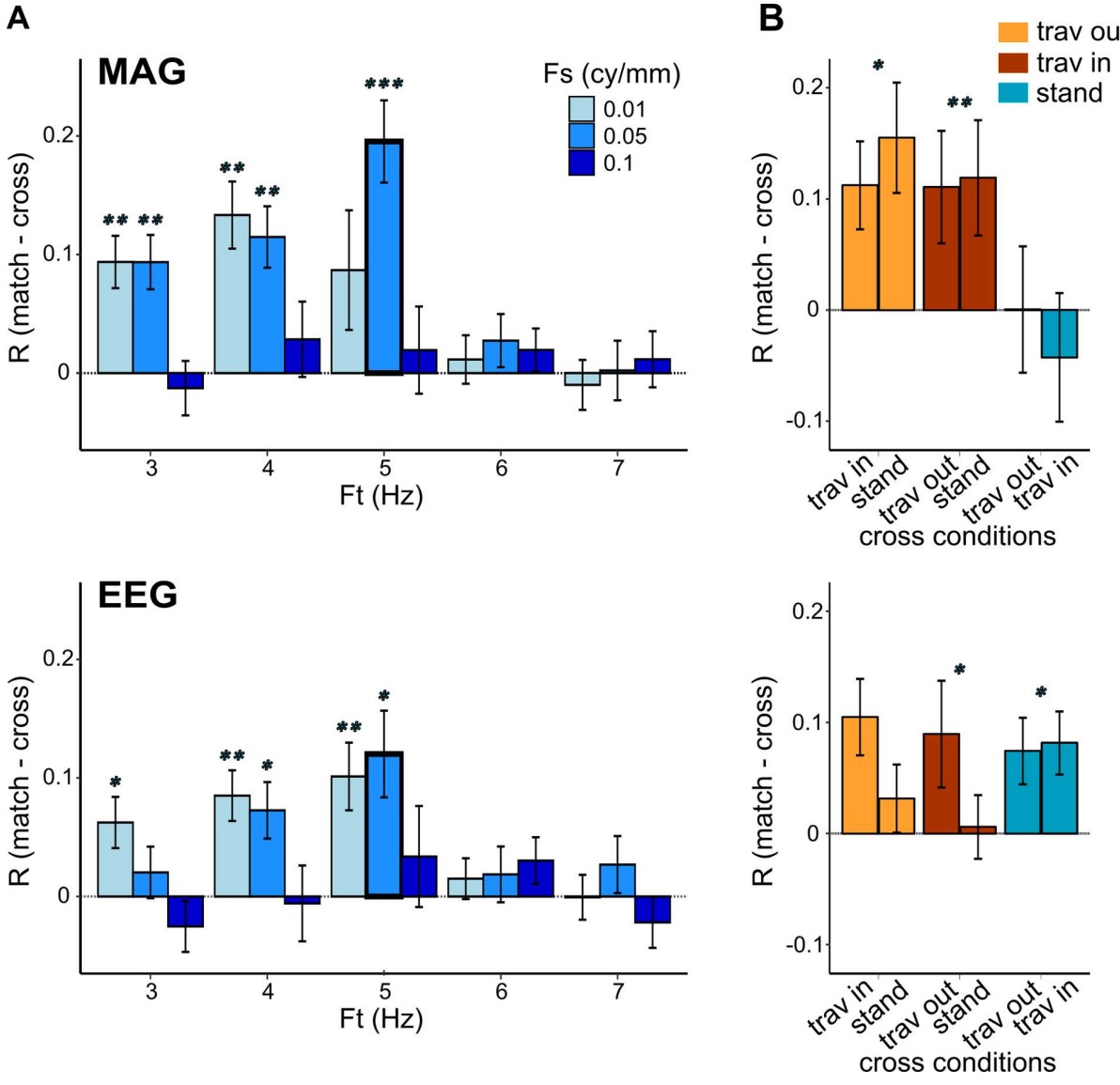

**Fig 5. The model is specific to the stimulus-induced traveling waves.** A. Using paired corrected t-tests, we compared the correlation coefficients between matched and crossed comparisons, for different values of temporal and spatial frequency. The bold outline bars correspond to the temporal (Ft) and spatial (Fs) frequency values used in the experimental conditions. Stars indicate significant p-values (*** p<0.001, ** p<0.01, * p<0.05). Error bars represent the Standard Error of the Mean (SEM). B. Paired t-tests on coefficients between matched and crossed comparisons were performed separately for each condition.

traveling out versus standing (MAG: 0.70±0.18, GRAD: 0.69±0.16, EEG: 0.80±0.10) and compared with traveling in versus standing (MAG: 0.70±0.18, GRAD: 0.69±0.16, EEG: 0.80±0.10). These results were obtained using a linear mixed model on the correlation coefficients, with comparisons and sensor type as factor, and participants as random effect. The main effect for comparisons was significant (F=21.55, p=7.0e-7) and there was a trend for sensor type (F=2.71, p=0.080). Post-hoc t-tests show significant differences between traveling out/in versus traveling out/standing (t=-5.43, p=1.25e-6) and traveling out/in versus traveling in/standing (t=-5.43, p=1.25e-6) and no significant difference between traveling out/standing versus traveling in/standing (t=-1.077, p=0.286).

## Discussion

Non-stationary, dynamical cortical states, and in particular oscillatory traveling waves, are increasingly becoming a topic of interest. Yet their non-invasive assessment within single cortical regions is made challenging due to methodological shortcomings. To assess local traveling waves in humans using non-invasive techniques, we developed a model-based neuroimaging approach that simulates wave propagation in V1 and projects this activity to the surface of the scalp to predict sensor signals. To validate the model, we used a MEG-EEG dataset recorded while participants viewed visual stimuli specifically designed to elicit traveling waves in V1. The method captures phase and amplitude relations across sensors at the stimulation frequency. Crucially, the model predictions were most accurate for the correct direction of traveling waves. The model was also sensitive to the temporal and spatial parameters of the traveling wave, as the predictions using the spatial and temporal frequencies corresponding to the actual visual stimulation were better at explaining the data.

Our models showed some specificity for condition (traveling in, traveling out, standing). For MEG sensors, the matched model was more accurate than the crossed model for traveling in and out conditions, but not for the standing condition. In other words, the standing data were equally well-explained by all three models: standing, traveling in, and traveling out. One factor is methodological: the predicted sensor patterns from the traveling wave models are relatively distinct from one another, whereas the predicted pattern from the standing wave model is relatively similar to the predictions from each of the traveling wave models. This explains why there is less specificity for the standing model than for the traveling models. This poorer discriminability, however, does not explain why the traveling in/out data were well predicted by their respective matched model, when compared to the standing model. An alternative reason might be that endogenous traveling waves might have also been generated by the standing visual stimuli, creating a mixture of a standing wave (that tracks the stimulus) and an endogenous traveling wave (that does not), a possibility that we did not implement in our models. Furthermore, the project was motivated by the possibility that the spontaneous response of V1 to any stimulus might manifest as a traveling wave of activity. Previous studies have shown that local traveling waves were triggered by a single visual event [22,24], potentially revealing structural network constraints that enable canonical operations of the visual system, such as divisive normalization [23,28]. One can also speculate that the standing stimuli generated traveling waves outside V1, within the visual hierarchy, which may have been partially captured by the traveling models, rather than the standing model.

We simultaneously acquired MEG and EEG data to evaluate the model's performance across these two techniques. Given that MEG and EEG detect sources in different orientations, differences between MEG and EEG outcomes are expected. We found three differences between the MEG and EEG results.

First, although the matched model predictions were more accurate than crossed model predictions for both MEG and EEG sensors, notable quantitative differences between MEG and EEG are observed. The correlations between the data and the predictions from non-matching models are significantly higher for EEG sensors compared to MEG sensors (Fig 4B). In other words, the EEG predictions are less specific than the MEG predictions. This can be explained by the fact that EEG signals are more affected by spatial mixing [29], which might diminish the granularity and therefore the specificity of the predicted signals.

Second, the predicted topographies qualitatively differ between MEG and EEG (Fig 4A). The model predicts traveling waves along the anterior-posterior axis for magnetometers, and along the lateral axis for EEG. While the magnetometer predictions are consistent with the magnetometers data, the EEG predictions are not consistent with the EEG data, where anterior-posterior phase shifts are observed. Previous studies mostly observed global traveling waves in EEG along the anterior-posterior axis [15,16,30], which might indicate that global traveling waves dominate the EEG data.

Last, further differences emerge when analyzing each condition separately. Contrary to MEG, the standing EEG data is better explained by the matched (standing) model compared to the crossed models (Fig 5B). It also appears that traveling in and out data are equally well-explained by their respective matched model (traveling in and out, respectively) as well as by the standing model. The origin of these differences is unclear, and it may be that each model (traveling and standing) explains different parts of the data.

The different patterns between MEG and EEG suggest that they have some degree of independence in their sensitivity to neural data. This implies that in principle, combining the two measures into an integrated analysis can lead to better characterization of cortical waves than either one alone. Alternatively, when using our approach one may need to determine which recording method (MEG or EEG) is most appropriate to compare two specific models. We thus recommend assessing the models' discriminability with each method by calculating the correlation between their predictions. This can be done using our open-access dataset and open-source toolbox prior data acquisition.

To sum up, this study presents a proof-of-concept for a model-based neuroimaging approach that enables the assessment of traveling waves in non-invasive recordings. We used a set of carefully controlled data with stimulus-induced traveling waves. By simulating activity only in V1, we were able to explain a substantial portion of the variance in the data (> 44%). The unexplained variance could stem from activity originating in other brain regions, as well as from intrinsic theta oscillations, which may be linked to attentional processes [4]. Future steps will involve adding other visual areas to the simulation. This model expansion raises new questions and will require testing assumptions on how waves propagate across multiple brain regions, such as determining the appropriate delays between different areas and including feedback connections [31]. A more complex next step in model development will involve modeling neural activity in non-retinotopic brain regions. Recent studies suggest that the direction of traveling waves may be constrained by anatomical structures [32,33]. One potential approach might use structural connectivity gradients to simulate traveling waves in non-retinotopic areas [34].

Overall, this newly developed toolbox aims to characterize traveling waves in spontaneous brain activity by contrasting models and the associated hypotheses regarding the propagation of neural oscillations. One could for instance use this toolbox to compare different computational hypotheses regarding the generation of MEG-EEG signals, as it was recently argued that phase shifts within sensor arrays can be explained by distinct sources [35,36]. Our method can be applied to assess whether sensor activity differs when a wave of activity propagates across the cortex or when spatially discrete dipole sources are oscillating (see [37] for further discussion). It can also address specific hypotheses regarding the functional link between traveling waves and cognition. Recent psychophysics results suggest that brain oscillations can propagate across retinotopic areas and in turn modulate performance periodically across the retinotopic space [19,20]. Our toolbox can be used to directly predict the link between sensor activity and behavioral performance across the retinotopic space. Extending the model to include non-retinotopic areas could further enable direct testing of hypotheses relating visual and higher-order regions, as it is for instance the case with attention. It was recently shown that traveling waves propagate between posterior and anterior regions in spatial attention tasks [37,38]. Our toolbox can be used in these tasks to assess potential travel paths and traveling waves' properties (e.g., speed). Together, this approach is a first step towards achieving an integrated view of brain functioning across time and space to understand the multiscale basis of cognition [39,40].

## Materials and methods

### Ethics statement

The study was performed at the CENIR imaging platform of the Paris Brain Institute (ICM), France, in accordance with the Declaration of Helsinki (2013) and approved by the French Ethics Committee on Human Research (RoFOCAC, #ID RCB 2017-A02787-46). All participants provided written informed consent.

### Participants

20 participants (11 female, age M = 28 years, SD = 6) took part in the study. All had normal or corrected-to-normal vision, and no known neurological or psychiatric disorder history. All participants were naive as to the purpose of the study, and were compensated for their participation. One participant was excluded a priori from the analysis due to excessive noise in the MEG signal. A total of 19 participants (10 female, age M = 28 years, SD = 5) were considered for the analyses.

## Anatomical and functional MRI acquisition

Anatomical MRI images were obtained using a 3-T Siemens Prisma MRI scanner with a T1-weighted (T1w) MRI sequence (FOV = 256 mm; 256 sagittal slices; 0.8 × 0.8 × 0.8 mm voxels, TR = 2400 ms, TE = 2.22 ms, flip angle = 9°, 7 min). Two additional multi-echo fast low-angle shot (FLASH) sequences (4 min each) with flip angles at 5° and 30° were also performed (FOV = 256 mm, 1 x 1 x 1 mm voxels, TR = 12 ms, TE = 1.52; 2.77; 4.02; 5.27; 6.52; 7.77; 9.02; 10.27 ms). Combined with the T1w images, they allow for a better delineation of the boundaries between brain and skull when calculating the head model than using the T1w images alone [41].

Functional MRI (FOV = 200 mm, 2 x 2 x 2 mm voxels, TR = 1000 ms, TE = 35 ms, flip angle = 62°) consisted of four 4.5-min long sequences during which visual stimuli were presented. Stimuli were black-and-white radial checkerboard patterns presented as a 90° wedge rotating clockwise or counterclockwise (to obtain phase maps) or a ring expanding or contracting (to obtain eccentricity [42–47]). They were generated using MATLAB (MathWorks) and the MGL toolbox [48] on a Macintosh computer, and presented using a ProPixx projector (VPixx Technologies, Saint-Bruno, QC, Canada; 120 Hz refresh rate, 1920x1080 pixels resolution, 24.3*13.8 degrees of visual angle, °VA). The display was projected on a screen located 120 cm away from the participant's eyes. Each cycle took 24 s. The radial checkerboard pattern contrast-reversed every 250 ms. Extra runs with counterclockwise wedges and expanding rings were added whenever time permitted (6 participants with 4 runs; 12 with 5 runs, which included 1 additional counterclockwise wedge block; and 10 with 6 runs, which included 1 additional counterclockwise wedge block and 1 additional expanding ring block).

## Retinotopy

Functional MRI data were preprocessed using fMRIPrep 20.2.1 version [49] (RRID:SCR_016216), a Nipype 1.5.1 version [50] (RRID:SCR_002502) based tool. Data were motion corrected, co-registered to the T1w with 6 degrees of freedom and slice-time corrected. A Bayesian pipeline developed by colleagues [51] was also implemented to allow accurate retinotopic mapping with a limited amount of fMRI data. This pipeline combines voxel-wise modeling of population receptive fields (pRF) with prior information from an anatomical atlas [51]. The pRF models were fit using the Matlab toolbox analyzePRF, based on a compressive spatial summation model [52,53]. The anatomical atlas was then fitted to the pRF data using the Neuropythy library [51].

## Stimuli for the MEG-EEG sessions

Matlab R2015B and Psychtoolbox 3.0.15 [54] were used for stimulus display. The stimuli were presented at a viewing distance of 78 cm on a gray screen with 1920*1080 pixels (79.5*44.5 cm; 32.5 x 18.3 °VA) using a PROPixx projector at 120 Hz refresh rate.

There were four conditions (Fig 1B): standing stimulus presented full-screen, a full-screen stimulus traveling from the periphery to the fovea (traveling in) or from the fovea to the periphery (traveling out), and a smaller stimulus (5°VA) around the fovea and traveling from the fovea to the periphery. This last condition was not analyzed in the current study.

To create each stimulus, a static carrier was multiplied by a contrast modulator (Fig 2A). This stimulus was designed to maximize the responses of V1 neurons, which increase with contrast [55]. The carrier was generated with bandpass-filtered Gaussian white noise with a spatial frequency that varied with eccentricity, following equation (Eq 1):

$$fs(e) = 1/(ae + b) \tag{1}$$

with $fs$ the preferred V1 spatial frequency (cycles/mm) for a given eccentricity e (°VA), a = 0.12, and b = 0.35 (values from [56]).

Binarization was then applied to maximize contrast. To avoid any adaptation effect to the static carrier, 10 different randomly created carriers were presented (one per block) in a pseudo-random order across participants (see one example carrier in Fig 2A). Within a block, the given carrier was presented in half of the trials in its original orientation and in the other half, rotated by 180°, in a pseudo-random order across conditions.

For the traveling stimulus, the contrast modulator was designed to elicit a traveling wave across retinotopic space. The contrast was modulated according to equation (Eq 2), with a spatial frequency F = 0.05 cycles/mm of cortex, a temporal frequency ω = 5 Hz and an initial phase shift $\varphi = \frac{\pi}{2}$. The wave amplitude and the offset (A = 0.5, c = 0.5, respectively) were chosen so that the contrast fluctuated between 0 (white screen) and 1 (black).

$$trav\_out(r,t) = A \sin \left( 2\pi F r - 2\pi \omega t + \varphi \right) + c \tag{2}$$

in which r corresponds to the radial distance from the center of the screen of a given pixel, after correction for cortical magnification.

The magnification inverse $M^{-1}$ scales with the eccentricity e of a given point on the screen (°VA) and follows equation (Eq 3) with $M_0$ = 23.07 mm/°VA, E2 = 0.75° VA (from Fig 9 in [57], parameters from [58]).

$$\mathrm{M}^{-1} = \mathrm{M_0}^{-1}(1 + \frac{1}{E2}\mathrm{e}) \tag{3}$$

The temporal frequency of the traveling stimulus was chosen to match the temporal frequency of endogenous traveling waves reported in invasive studies in humans [11,18]. The spatial speed (0.05 cycles/mm at 5 Hz corresponding to a wave propagating at 0.1 m/s) also matched the empirically observed velocity of local traveling waves (0.1-0.8 m/s, from [20,31]).

For the standing stimulus, the contrast was modulated using equation (Eq 4) after adjusting for cortical magnification, with the same parameters as the traveling stimulus.

$$stand(r,t) = A \sin(2\pi F r) \cos \left( 2\pi \omega t + \varphi \right) + c \tag{4}$$

The traveling in condition followed Eq 5, with the same parameters as Eq 2.

$$trav\_in(r,t) = A \sin \left( 2\pi F r + 2\pi \omega t + \varphi \right) + c \tag{5}$$

The stimuli were finally obtained by multiplying the contrast modulator with the carrier, after rescaling between 0 (white) and 1 (black) as follows:

$$stim(r,t) \ = \ (carrier(r).mod(r,t) \ + \ 1)/2 \tag{6}$$

with mod being either *trav_out*, *stand* or *trav_in,* as defined in Equations 2, 4 and 5, and r the radial distance from the center of the screen of a given pixel. Snapshots of the stimuli are shown in Fig 2B, with the corresponding simulated cortical activity.

## Experimental procedure

Participants were asked to maintain fixation on a red dot at the center of the screen (0.05 °VA of diameter; Fig 1B). A trial consisted of a visual stimulus presented for 2 seconds, followed by a 2-second interval with a gray screen (mean luminance) and the fixation dot. A run consisted of 8 trials, with 2 repetitions for each condition (traveling out, traveling in, standing and traveling out_fov), randomly interleaved. A block was composed of 8 runs, and 10 blocks (~7min each) were performed (160 trials total per condition) within one recording session (~1h30 with breaks, within a 4h-session including setting up MEG and EEG). Two 3-min segments of resting-state, one with eyes opened and one with eyes closed, were additionally recorded both at the beginning and at the end of the session.

To maintain participants' attention during stimulus presentation, at the end of each run they were asked to indicate whether the color of the fixation dot changed briefly from red to yellow (probability of 25%) by pressing a key (keys were counterbalanced across participants, i.e., right key for yes, and left key for no, or vice versa). In 25% of the runs, the color

change occurred once during one of the 8 trials (random), at a random time between 0.5 to 1.8 s from stimulus onset and lasted for 150 ms. There was no color change in 75% of the runs. Participants had 4 s maximum at the end of the run to respond otherwise the answer was not registered. Once participants responded, they were invited to rest their eyes before the next run. Note that because of the low number of trials in which a color change occurred (20 out of 640 trials), their random timing, and their brief duration, potential effects on the MEG-EEG signals were considered reasonably minimized during trial averaging and thus included in all analyses. After 10 s, a sound (0.5 s sinusoidal signal with 5 ms fade-in and fade-out, at 1 kHz) warned them that the next run would start 2 s later. Participants' sensitivity was assessed with d', computed by subtracting the z-transformed false alarm rate (FA) from the z-transformed hit rate (H). The average d' was $1.26 \pm 0.31$ (H = 52 ± 5%, FA = 21 ± 5%), indicating that participants were performing the task well above chance and successfully fixated the center dot, which is critical when interested in retinotopic representations.

## MEG-EEG recording

Electromagnetic brain activity was simultaneously recorded with EEG and MEG (MEG-EEG) in a magnetically shielded room. MEG was collected using a whole-head Elekta Neuromag TRIUX MEG system (Neuromag Elekta LTD, Helsinki) in upright position, a system equipped with 102 triple sensor elements (one magnetometer, MAG, and two orthogonal planar gradiometers, GRAD). EEG was recorded using an Easycap EEG cap compatible with MEG recordings (BrainCap-MEG, 74 electrodes, 10–10 system).

The MEG-EEG signal was recorded at 1 kHz sampling frequency with a low-pass filter of 330 Hz. The EEG signal was further recorded with a high-pass filter at 0.01 Hz. Horizontal and vertical electro-oculograms (EOG) were collected using two electrodes near the lateral canthi of both eyes and two electrodes above and below the dominant eye. Electrocardiogram (ECG) was recorded with two electrodes placed on the right collarbone and lower left side of the abdomen. The ground electrode for the EEG was placed on the left scapula and the signal was referenced to connected electrodes on the left and right mastoids.

Participants' head position was measured before each block using four head position coils (HPI) placed on the EEG cap over the frontal and parietal areas. Anatomical landmarks (nasion and the left and right pre-auricular areas) were used as fiducial points. The HPI, fiducial points, and EEG electrodes were digitalized using a 3D digitizer (Polhemus Incorporated, VT, USA) to help with the co-registration of MEG-EEG data to the anatomical MRI.

## MEG-EEG pre-processing

Noisy MEG sensors were excluded after visual inspection of raw data. Signal space separation (SSS) was then applied to the raw continuous data to decrease the impact of external noise [59]. The head position measured at the beginning of each block was realigned relative to the block coordinates closest to the average head position across blocks. SSS correction, head movement compensation, and noisy MEG sensor interpolation were applied using the MaxFilter Software (Elekta Neuromag). Using the MNE-Python suite [60], noisy EEG sensors were interpolated by spherical splines. We did not reject trials based on eye blink or saccade since it would represent only a small portion of the whole 2 s-trial. Yet, ocular and cardiac MEG-EEG artifacts were reduced using ICA decomposition in the following manner. First, MEG-EEG data were aligned to the detected ECG and EOG peaks. ICA was then performed and components capturing ocular and cardiac artifacts were identified separately for EEG and MEG recordings. The correction consisted in rejecting ICA components most correlated with the ECG and EOG signals [61]. The rejection criterion was defined by a Pearson correlation score between MEG-EEG data and the ECG- and EOG-signals with an adaptive z-scoring (threshold = 3; same as in [62]). All outcomes were verified by visual inspection. Lastly, the EEG signal was re-referenced to the common average.

The measured MEG-EEG data were filtered between 1 Hz and 45 Hz using zero-phase FIR filters (recommended by [63], and down-sampled to 200 Hz. For each stimulus condition (Fig 1A), the data were epoched in 3.5 s-segments, from -1 s to 2.5 s relative to stimulus onset. Trials exceeding a certain amplitude threshold were automatically rejected

(GRAD: 4e-10 T/m; MAG: 6e-12 T; EEG = 1e-4 V; values chosen based on MNE-python recommendation, and average data quality). There were, on average across participants, 142 epochs left per condition (SD = 7, i.e., 11 ± 4% of rejected trials) after rejection. These epochs were then averaged for each condition, yielding three 3.5-s time series per sensor, one each for traveling out, traveling in, and standing waves.

### Head model

Anatomical MRIs were segmented with the FreeSurfer image analysis suite (http://surfer.nmr.mgh.harvard.edu). A 3-layer boundary element model (BEM) surface was generated to constrain the forward model. Individual head models (4098 octahedrons per hemisphere, 4.9 mm spacing) were computed using the individual BEM model constrained by the anatomical MRI. The anatomical MRI and MEG-EEG sensors were manually realigned with a coregistration procedure based on digitized anatomical landmarks.

### Modeling traveling and standing waves

All models and analyses were performed in Python with the MNE-Python toolbox [60] and custom scripts (see **Code and data availability** section and [64]). Traveling and standing waves were modeled in V1 based on the visual stimuli used in the MEG-EEG experiment (Fig 2B). Using the eccentricity map derived from the retinotopy analysis, we mapped the space- and time-varying contrast of the stimuli onto the time-varying values of each voxel of the source space in V1 (Fig 1). These fluctuations therefore followed Eq 3, 5 and 6 and were converted from units of contrast to units of equivalent current dipole with the assumption of 10 nA.m per unit of contrast. This generated a peak of 10 nA.m (same value as in [65,66]). The simulated source time series were then projected onto the MEG and EEG sensors, using the head model.

### Comparison between predicted and measured data

The across-trial averaged time series were used to compare the model to the data. To quantify model accuracy, we compared the phase and amplitude between predicted and measured data at 5 Hz, the stimulus frequency. The complex-valued 5-Hz Fourier coefficients were calculated using the Fast Fourier transform (FFT) of the time series from 0 (stimulus onset) to 2 s, separately for predicted and measured data and for each sensor. We then compared measured and predicted data by correlating the complex values across sensors, separately for each sensor type (MAG, GRAD, EEG), after concatenating traveling out, traveling in and standing complex values. The Pearson correlation coefficient of complex values was calculated using the *corrcoef* function from the *numpy* package. The absolute value of these complex coefficients is reported here (R). The concatenation produces vectors with length 3 times the number of sensors. So, for example, for the magnetometers, the correlation coefficient was computed between two pairs of complex vectors, one for data and one for model predictions, each with length 306 (3 times the number of magnetometers). The reason for concatenation across conditions is that correlation of complex values is insensitive to a global phase shift; if each condition were correlated separately, this would allow for three degrees of freedom in the phase rather than one, increasing the chance of overfitting.

All analyses were performed separately for the three sensor types (MAG, GRAD and EEG). In the main figures, we only report MAG and EEG. Gradiometers are reported in S1 and S2 Figs.

### Statistical analysis

To assess whether the model predictions correlate with the data, we performed permutation-based statistics on the correlation coefficients calculated from all concatenated conditions, both at the individual and group level. For each participant, we shuffled the predicted data across sensors and correlated it with the measured data. This procedure was repeated 5,000 times to obtain an individual-participant surrogate distribution of correlation coefficients. The distribution was Fisher-Z transformed to obtain a normal distribution. The p-value was calculated as the corresponding percentile of

the empirical coefficient in the surrogate distribution. At the group level, a surrogate distribution of Fisher-Z-transformed coefficients was generated by randomly selecting a sample from each individual distribution and averaging samples across participants (10,000 repetitions). The group-level p-value corresponded to the percentile of the group-averaged coefficient in the surrogate distribution.

To test whether the correlation coefficients differed between conditions, we computed a null model by concatenating all 6 possible combinations for crossed conditions (e.g., traveling out data vs. traveling in model, traveling out data vs. standing model…etc.). We used linear mixed models on correlation coefficients with condition (matched or crossed) as predictor and participants as random effect, using R 4.2.2 (R Core Team, 2022). Effect sizes were calculated using Cohen's q, a measure to compare correlations coefficients, defined as the difference between Fisher-Z-transformed group-averaged coefficients. Small, medium and large size effects are defined as q = 0.1, 0.3, and 0.5, respectively [67].

## Supporting information

**S1 Fig. Power Spectrum Density (PSD) was calculated on single-trial data for all conditions together (traveling out, traveling in and standing) and each sensor type separately (MAG, GRAD, EEG).** The group-averaged PSD is averaged across sensors (black line: mean, shaded area: SEM) and plotted for a selected sensor (blue). The 5-Hz component and its harmonics are clearly visible. The inset topomaps show the PSD at 5 Hz for each sensor type. (TIFF)

**S2 Fig. The analysis conducted in Fig 4 in MAG and EEG were also performed in GRAD.** Same legend as for Figs 4 and 5. (TIFF)

**S1 Video. Traveling wave activity simulated in the primary visual cortex of one individual, for each condition (left: traveling out, middle: traveling in, right: standing).** (GIF)

**S2 Video. Projection of the simulated activity on magnetometers (top) and EEG sensors (bottom), for each condition (left: traveling out, middle: traveling in, right: standing).** (GIF)

## Author contributions

**Conceptualization:** Laetitia Grabot, Jonathan Winawer, David J. Heeger, Laura Dugué.

**Data curation:** Laetitia Grabot.

**Formal analysis:** Laetitia Grabot.

**Funding acquisition:** Laura Dugué.

**Investigation:** Laetitia Grabot, Laura Dugué.

**Methodology:** Laetitia Grabot, Garance Merholz, Jonathan Winawer, David J. Heeger, Laura Dugué.

**Project administration:** Laura Dugué.

**Supervision:** Laura Dugué.

**Validation:** Laetitia Grabot, Jonathan Winawer, David J. Heeger, Laura Dugué.

**Visualization:** Laetitia Grabot, Laura Dugué.

**Writing – original draft:** Laetitia Grabot.

**Writing – review & editing:** Laetitia Grabot, Garance Merholz, Jonathan Winawer, David J. Heeger, Laura Dugué.

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
