## [Decision Letter · Decision Letter 0]

13 Dec 2024

PCOMPBIOL-D-24-01820

Traveling Waves in the Human Visual Cortex: an MEG-EEG Model-Based Approach

PLOS Computational Biology

Dear Dr. Grabot,

Thank you for submitting your manuscript to PLOS Computational Biology. After careful consideration, we feel that it has merit but does not fully meet PLOS Computational Biology's publication criteria as it currently stands. Therefore, we invite you to submit a revised version of the manuscript that addresses the points raised during the review process.

Please submit your revised manuscript within 30 days Feb 12 2025 11:59PM. If you will need more time than this to complete your revisions, please reply to this message or contact the journal office at ploscompbiol@plos.org. Please include the following items when submitting your revised manuscript:

We look forward to receiving your revised manuscript.

Kind regards,

Alain Nogaret, PhD

Academic Editor

PLOS Computational Biology

Joseph Ayers

Section Editor

PLOS Computational Biology

**Additional Editor Comments:**

All three reviewers have deemed the work interesting, original and well-written. They have also made suggestions for minor improvements and for broadening the discussion that you will want to consider.

**Journal Requirements:**

3) We notice that your supplementary Figures are included in the manuscript file. Please remove them and upload them with the file type 'Supporting Information'. Please ensure that each Supporting Information file has a legend listed in the manuscript after the references list.

4) Please ensure that the funders and grant numbers match between the Financial Disclosure field and the Funding Information tab in your submission form. Note that the funders must be provided in the same order in both places as well.

If you did not receive any funding for this study, please simply state: u201cThe authors received no specific funding for this work.u201

**Reviewers' comments:**

Reviewer's Responses to Questions

**Comments to the Authors:**

Reviewer #1: It is an interesting study; the reviewer has some comments.

1. Please consider further exploring why the standing model does not show greater specificity. Alternatively, discuss in more detail the implications of standing stimuli potentially generating endogenous traveling waves.

2. The study highlights differences in the specificity and predicted topographies for MEG and EEG, with EEG predictions showing less granularity and specificity. Please elaborate on the practical implications of these differences.

3. The study is currently limited to simulating activity in V1. Provide a brief outline of how the model could be expanded to include multiple areas.

4. The study briefly mentions that the tool could test hypotheses about neural oscillation propagation and the generation of MEG-EEG signals. Please strengthen this point by providing concrete examples of such hypotheses and how this tool could be applied.

5. The model explains over 44% of the variance in the data, which is promising but leaves a significant portion unexplained. Please discuss the potential sources of unexplained variance.

6. Please expand on the observed differences between MEG and EEG predictions in the Discussion section. Specifically, consider elaborating on the practical implications of these differences for integrating MEG and EEG data in future studies.

Reviewer #2: In the present work, Grabot and colleagues aim at developing a model-based neuroimaging approach to assess traveling waves in non-invasive recordings. They compared the model's performances to MEG-EEG signals obtained during controlled experimental conditions. The authors were able to explain 44% of the variance of data.

The methodology is rigorous and the results well presented, with useful and detailed figures.

I have only a few minor concerns that I would ask the authors to address to improve the manuscript's clarity.

INTORDUCTION

- Before starting describing the two-part model, I would add a brief description of the EEG and MEG methodologies, especially focusing on the fact that spacial discrimination is not their main strenght.

-I would also suggest to better define "standing" and "travelling" waves.

RESULTS

- The authors say that they can explain around 44% of the variance of the data: I would add a comparison with other models or a sort of comparator to give an effect size of predictive

performance (mild-moderate-great).

DISCUSSION

- Ath the end of the discussion session, I would reprase the conclusive sentence "This approach aligns with the advocated need for an integrated view of brain functioning across time and space to understand the multiscale basis of cognition". Given that the authors claim it's the first work of the kind, I would not overstate the implications of their novel approach.

Reviewer #3: This is a very intriguing, novel, well-designed, and well-written work. I have only a few minor comments.

Please provide full information about the funding: Initials of the authors who received each award; URL of each funder website; Did the sponsors or funders play any role in the study design, data collection and analysis, decision to publish, or preparation of the manuscript?

Please indicate a possible significance of the results in cognition. The authors only briefly mention this matter in one sentence. I'd like the Authors to expand this fragment.

Please provide explanations of the abbreviations used in the figures. E.g. in fig. 1.: 'VM' and 'HM'.

Experimental procedure

Could you please add the mean total time of the experimental session? A figure presenting the experimental protocol would also help in visualizing the details of the experimental procedure.

Since the color change to maintain participants' attention occurred during the M/EEG epoch - were these trials excluded from the analysis?

MEG-EEG pre-processing

How were the blinks during the stimulus presentation handled? Usually, before the ICA, the stimuli during which an eye blink occurred are identified, and then these trials are excluded from the analysis to prevent analyzing trials where the participants did not see the stimulus.

Comparison between predicted and measured data

I think the information provided in this section (or at least the first part) should be moved to the 'MEG-EEG pre-processing' section.

Results

Some of the information is repeated after other sections.

**Have the authors made all data and (if applicable) computational code underlying the findings in their manuscript fully available?**

Reviewer #1: Yes

Reviewer #2: Yes

Reviewer #3: Yes

PLOS authors have the option to publish the peer review history of their article (what does this mean? ). If published, this will include your full peer review and any attached files.

**Do you want your identity to be public for this peer review?** For information about this choice, including consent withdrawal, please see our Privacy Policy .

Reviewer #1: No

Reviewer #2: No

Reviewer #3: No

**Figure resubmission:**
---

## [Editor Report · Decision Letter 1]

27 Mar 2025

Dear Dr Grabot,

We are pleased to inform you that your manuscript 'Traveling Waves in the Human Visual Cortex: an MEG-EEG Model-Based Approach' has been provisionally accepted for publication in PLOS Computational Biology.

Best regards,

Alain Nogaret, PhD

Academic Editor

PLOS Computational Biology

Joseph Ayers

Section Editor

PLOS Computational Biology

Daniele Marinazzo

Section Editor

PLOS Computational Biology

Having carefully checked the changes made in the manuscript in response to reviewers queries, it is clear that the additional details provided on the method and implications of the findings have met these queries.

---

## [Editor Report · Acceptance letter]

PCOMPBIOL-D-24-01820R1

Traveling Waves in the Human Visual Cortex: an MEG-EEG Model-Based Approach

Dear Dr Grabot,

I am pleased to inform you that your manuscript has been formally accepted for publication in PLOS Computational Biology. Your manuscript is now with our production department and you will be notified of the publication date in due course.

With kind regards,

Anita Estes
